# Learning Fused State Representations for Control from Multi-View Observations

## Abstract

In visual control tasks, leveraging observations from multiple views enables Reinforcement Learning (RL) agents to perceive the environment more effectively. However, while multi-view observations enrich decision-making information, they also increase the dimension of observation space and introduce more redundant information. Thus, how to learn compact and task-relevant representations from multi-view observations for downstream RL tasks remains a challenge. In this paper, we propose a Multi-view Fusion State for Control (MFSC), which integrates a self-attention mechanism with bisimulation metric learning to fuse task-relevant representations from multi-view observations. To foster more compact fused representations, we also incorporate a mask-based latent reconstruction auxiliary task to learn cross-view information. Additionally, this mechanism of mask and reconstruction can enpower the model with the ability to handle missing views by learning an additional mask tokens. We conducted extensive experiments on the Meta-World and Pybullet benchmarks, and the results demonstrate that our proposed method outperforms other multi-view RL algorithms and effectively aggregates task-relevant details from multi-view observations, coordinating attention across different views.

## 1 Introduction

In robotic manipulation tasks, acquiring accurate 3D scene information including understanding the target position, orientation, occlusions, and the stacking relationships among objects in complex environments, is crucial for effective grasping and interaction with objects. However, utilizing 3D inputs poses significant challenges, including increased computational complexity and the difficulty of extracting spatial information effectively from such data. In order to alleviate this problem, recent researches (Li et al. (2019), Chen et al. (2021), Jangir et al. (2022), Hwang et al. (2023), Seo et al. (2023b)) on Multi-View Reinforcement Learning (MVRL), which leverages 2D observations from multiple perspective cameras to enhance perception and understanding of spatial relationships, effectively mitigates the complexity of 3D input. However, while multi-view observations enhance the agent's comprehension of the environment and improve decision-making, they also increase the complexity of learning effective multi-view representations. We have summarized two challenges that arise in multi-view representation learning: 1) **Higher data dimensions and more redundant information** The multi-view observations composed of multiple high-dimensional images significantly increase the dimension of data. These high-dimensional observations not only increase computational costs but may also introduce substantial amounts of irrelevant or redundant information, such as shadows, thereby diminishing learning efficiency (Kaiser et al. (2019), Lake et al. (2017)); 2) **Informative aggregation of representation from various views.** During the task process, the amount of relevant information provided by different perspectives varies. Thus, excessive reliance on a single viewpoint impedes a comprehensive understanding of the environment and undermines robustness in scenarios where certain views are absent (Hwang et al. (2023)).

To address *Challenge 1*, previous studies of co-regularized multi-view learning (MVL) combined with deep learning techniques, has made significant progress, especially in utilizing complementary information from multi-modal data or features. Related research includes multi-view generative models (Wu & Goodman (2018), Sutter et al. (2020), Shi et al. (2019), Hwang et al. (2021)), multi-view auto-encoders (Wang et al. (2019)), and applications of deep belief networks (Kang & Choi (2011)). However, these methods often face difficulties in real-world control tasks, as they tend

to overemphasize task-irrelevant details, making it challenging to effectively extract and fuse the critical state representations necessary for control tasks. In contrast, *Challenge 2* emphasizes the importance of effectively aggregating information from diverse views, which is pivotal for improving learning performance. Each view contributes unique and complementary insights into the task, and appropriately leveraging these contributions is crucial. However, previous works have often introduced an inductive bias that multi-view information is considered equivalent, or that one view is assumed to provide more information by default. For example, (Akinola et al. (2020)) obtains the fused representation of multi-view observations by merely concatenating the representations from each individual view. While some works, such as Jangir et al. (2022), measure the significance of information from different perspectives through cross-view attention mechanisms, the computational complexity increases quadratically with the number of views, and it cannot guarantee that the aggregated information is task-relevant. To ensure that multi-view fusion is maximally task-relevant, it is imperative to closely align the integration process with the specific objectives of the task. By doing so, we can more comprehensively capture the underlying structures and patterns, thereby facilitating enhanced control.

To learn compact and task-relevant representations from multi-view observations, we propose a novel architecture—Multi-view Fusion State for Control (MFSC). First, we consider the observed image of each view in MVRL as a token in NLP. Inspired by Bert (Devlin (2018)) and ViT (Dosovitskiy (2020)), the [class] token, which can be viewed as the summarization of the whole sentence or picture, is used as the learnable fusion state representations from multi-view observations. This additional learnable fusion representation can prevent the model from over-focusing on observations from a single perspective to balance the aggregation of information represented in various views. Simultaneously, we incorporate bisimulation principles by integrating reward signals and dynamic differences into the fused state representation to capture task-relevant details. Additionally, this architecture employs a masking strategy based on cross-view consistency to encourage the learning of consistent information across views. This masking strategy encourages the model to learn consistent information across viewpoints by masking information in certain views. A key feature of our method is the reconstruction of the masked observations, ensuring that their latent features match those of the original branch in the latent representation space rather than the pixel space.

As a multi-view fusion state representation learning module, MFSC can be seamlessly integrated into any existing downstream reinforcement learning framework, enhancing the agent's understanding of the environment. We evaluated MFSC on the Meta-World (Yu et al. (2020)) and Pybullet (Coumans & Bai (2022)) benchmarks with the following analyses. First, we assessed MFSC's performance in MVRL and compared it against other methods on Meta-World. Second, we tested it on high-dimensional control problems using Pybullet, showing that our algorithm effectively captures task-relevant information. Third, we evaluated MFSC's robustness to missing views. Finally, we visualized MFSC's attention both across and within views. Our project code is publicly available at `https://anonymous.4open.science/r/MFSC-F57B`.

## 2 RELATED WORK

### 2.1 MULTI-VIEW LEARNING

Multi-view learning is typically divided into three main strategies (Sun (2013), Zhao et al. (2017)): co-training, multi-kernel fusion, and co-regularization (Guo & Wu (2019)). The co-training approach utilizes labeled data to iteratively train classifiers for each view and labels unlabeled data based on the predictions of these classifiers (Kumar & Daumé (2011), Ma et al. (2017)). Kernel methods combine the kernel matrices from different views to learn a global representation based on the fused kernel (De Sa et al. (2010), Li et al. (2015)). Co-regularization methods add regularization terms to encourage consistency among data from different views. Traditional co-regularization techniques include (i) methods based on Canonical Correlation Analysis (CCA) (Vía et al. (2007)), Sindhwani & Rosenberg (2008), Guo & Xiao (2012), Andrew et al. (2013), Jin et al. (2014), Guo & Wu (2019), and (ii) Linear Discriminant Analysis (LDA) methods that require labeled data (Jin et al. (2014)). With the development of deep generative models, co-regularization-based multi-view learning has made significant progress (Wu & Goodman (2018), Sutter et al. (2020), Shi et al. (2019), Hwang et al. (2021)). Specifically, multi-view generative models jointly train data from different views and use regularization mechanisms to ensure that the latent representations of each view share

a consistent and complementary information space. In vision-based control tasks, directly applying multi-view learning often results in low efficiency in learning state representations (Hwang et al. (2023)), which negatively impacts subsequent reinforcement learning (RL) algorithms. We propose a co-regularization training approach that leverages the reward and state transition mechanisms in RL, combined with masked latent space reconstruction, to learn an effective state fusion representation from pixel-based multi-view observations.

## 2.2 REINFORCEMENT LEARNING FROM MULTI-VIEW OBSERVATIONS

Effective state representation learning in MVRL aims to construct a mapping function that transforms rich, high-dimensional multi-view observations into a compact latent space. Recent research has explored various methods for representation learning in MVRL. Li et al. (2019) proposed a multi-view RL algorithm based on the Variational Autoencoder architecture (Kingma (2013)). This model discards the notion of a joint state by minimizing the Euclidean distance between the state encoded from the primary view and the states encoded from other views, assuming the first view is always available as the primary view. Chen et al. (2021) learns 3D visual keypoints through 3D reconstruction from multiple third-person views, however it requires additional information such as camera calibration parameters. Jangir et al. (2022) addresses MVRL with egocentric and third-person images, using cross-view attention to aggregate representations without calibration. While it can extend to multiple views, the computational cost grows quadratically with the number of views, limiting efficiency. Hwang et al. (2023) explored information-theoretic methods to capture the underlying state space model from multi-view observation sequences, addressing the problem of missing views. Seo et al. (2023b) employs a multi-view masked autoencoder to reconstruct the pixels of randomly masked viewpoints. Following this, a world model is learned based on the representations from the autoencoder. Our work seeks to explore the use of reward signals in RL to facilitate the learning of fused state representations. Building on the Transformer-Encoder architecture (Vaswani (2017)), our approach employs reward-guided bisimulation to ensure that the fused state representations capture sufficient information. Additionally, we enhance the model's representation learning capabilities by leveraging masked latent space prediction to exploit inter-view correlations, enabling more effective learning of fused state representations from multi-view observations.

## 3 MULTI-VIEW MARKOV DECISION PROCESSES

To enable an agent to adapt to multi-view observations, we extend the concept of the Markov Decision Process (MDP) to a Multi-View Markov Decision Process(MV-MDP), which is defined by the following tuple $\langle \mathcal{S}, \mathcal{A}, \vec{\mathcal{O}}, \mathcal{P}, \Omega, \mathcal{R}, \gamma, p_0 \rangle$. $\mathcal{S}$ represents the set of ground-truth states $s$ in the environment, $\mathcal{A}$ is a set of actions $a$, $\vec{\mathcal{O}} = \{\mathcal{O}^v\}_{v=1}^V$ represents the set of $V$ observations. With the assumption that each multi-view observation $\vec{o} \in \vec{\mathcal{O}}$ uniquely determines its generating state $s \in S$, we can obtain the latent state regarding its multi-view observation by a projection function $\phi(\vec{o}) : \vec{\mathcal{O}} \to S$. Therefore, $s$ and $\phi(\vec{o})$ can be used interchangeably. $\mathcal{P}(s'|s,a) = \Pr(s_{t+1} = s'|s_t = s, a_t = a)$ is the transition dynamics distribution. The corresponding transition function under the multi-view observation space is defined $\vec{o}' \sim \hat{\mathcal{P}}(\vec{o}'|\vec{o},a)$, where $\hat{\mathcal{P}}(\vec{o}'|\vec{o},a) = \Omega(\vec{o}'|s')\mathcal{P}(s'|s,a)$ and $\Omega(\vec{o}|s) = \prod_{v=1}^V \Pr(o_t^v = o^v|s_t = s)$ is the joint observation probability distribution. $\mathcal{R}(s,a) \in \mathbb{R}$ is the immediate reward function for taking action $a$ at state $s$, $\gamma \in [0, 1]$ is the discount factor and $p_0(s) = \Pr(s_0 = s)$ is the starting state distribution at timestep 0. The goal of the agent is to find the optimal policy $\pi(a|s)$ to maximize the expected reward: $E_{s_0,a_0,...}[\sum_{t=0}^\infty \gamma^t r(s_t, a_t)]$. In addition, if contextual information is required, we can approximate stacked pixel images as observations.

## 4 ANALYSIS ON SAMPLE-EFFICIENCY OF MULTI-VIEW REINFORCEMENT LEARNING

In MVRL, different views of observations may contain redundant or irrelevant information. Instead of solving the original multi-view RL problem, we can learn a *summarized MDP* to simplify the problem. In this *summarized MDP*, the actions remain unchanged, while the dynamics and reward function are parameterized by the summarizations rather than raw multi-view observations. We formalize our intuition into the following:

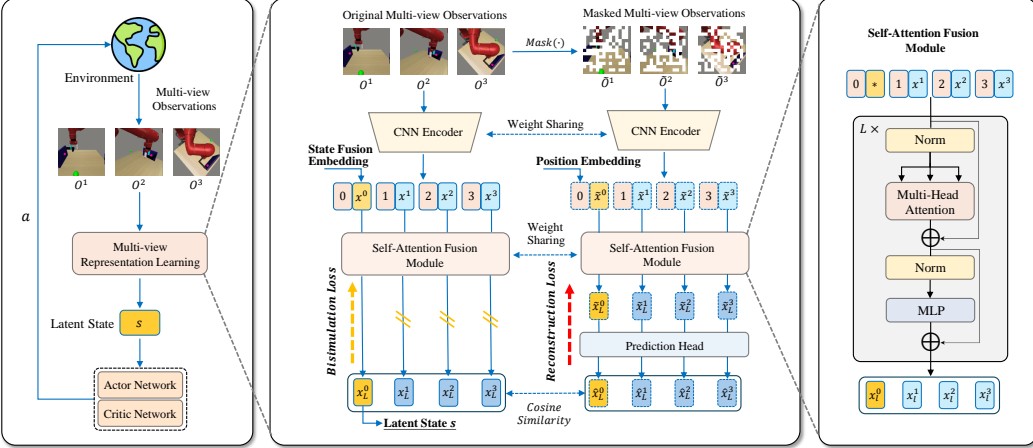

Figure 1: Framework of MFSC. (a) The left part illustrates the process of MVRL, where the agent receives observations from multiple views, learns a fused latent state, and interacts with the environment through an actor-critic framework. (b) The middle part provides a detailed overview of the MFSC architecture. Each view is encoded into a latent embedding via a Convolutional Neural Network (CNN), followed by state fusion using the Self-Attention Fusion Module. Metric learning is guided by bisimulation loss, and a mask-based self-supervised auxiliary task is employed to enhance the model's cross-view learning capabilities. (c) The right part presents the inner workings of the Self-Attention Fusion Module, which integrates embeddings from different views through attention mechanisms to produce a unified state representation.

**Assumption 1.** *There exists a set $\mathcal{Z}$ where $|\mathcal{Z}| \ll |\mathcal{O}^1 \times \mathcal{O}^2 \times \cdots \times \mathcal{O}^k|$, and $\varepsilon > 0$, such that the summarized MDP $\langle \mathcal{Z}, \mathcal{A}, \mathcal{P}, \Omega, \mathcal{R}, \gamma, p_0 \rangle$ satisfies: for every $\vec{o} = (o_1, o_2, ..., o_k) \in \mathcal{O}^1 \times \mathcal{O}^2 \times \cdots \times \mathcal{O}^k$, there exists a $z \in \mathcal{Z}$ satisfying $|V^*(\vec{o}) - V^*(z)| \leq \varepsilon$.*

Based on **Assumption** 1, we can abstract the space of multi-view observations into a much more compact space of summarizations, retaining only the features relevant to action selection. In practice, summarizations can be generated by aggregating the different multi-view observations. In the following section, we will present the use of bisimulation metrics to abstract the space of multi-view observations, offering strong theoretical guarantees.

## 4.1 ABSTRACTING MULTI-VIEW OBSERVATIONS WITH BISIMULATION METRICS

In single-view RL tasks, bisimulation metric learning has been proven to be an effective method for acquiring robust state representations Zhang et al. (2021); Zang et al. (2022; 2023); Sun et al. (2024). In this paper, we extend the task setting from a single view to multi views, and demonstrate that employing bisimulation metrics for representation learning can similarly enhance both the theoretical and empirical performance of standard RL algorithms in MVRL.

Formally, as described in Castro et al. (2021) we define the bisimulation metric for policy $\pi$ on a multi-view setting as:

$$\mathcal{F}^\pi G^\pi(\vec{o}_i, \vec{o}_j) = |r^\pi_{\vec{o}_i} - r^\pi_{\vec{o}_j}| + \gamma \mathbb{E}_{\vec{o}'_i \sim \mathcal{P}^\pi_{\vec{o}_i}, \vec{o}'_j \sim \mathcal{P}^\pi_{\vec{o}_j}}[G^\pi(\vec{o}'_i, \vec{o}'_j)], \tag{1}$$

where $\vec{o}_i, \vec{o}_j \in \mathcal{O}^1 \times \mathcal{O}^2 \times \cdots \times \mathcal{O}^k$. Zhang et al. (2021) suggested that learning an approximate value of the bisimulation metric in the embedding space can be more practical than utilizing the true bisimulation metric. Similarly, we propose learning an aggregator $\phi : \mathcal{O}^1 \times \mathcal{O}^2 \times \cdots \times \mathcal{O}^k \to \mathbb{R}^d$:

$$\mathcal{F}^\pi G^\pi(\phi(\vec{o}_i), \phi(\vec{o}_j)) = |r^\pi_{\phi(\vec{o}_i)} - r^\pi_{\phi(\vec{o}_j)}| + \gamma \mathbb{E}_{\vec{o}'_i \sim \mathcal{P}^\pi_{\vec{o}_i}, \vec{o}'_j \sim \mathcal{P}^\pi_{\vec{o}_j}}[G^\pi(\phi(\vec{o}'_i), \phi(\vec{o}'_j))], \tag{2}$$

where the operator $\mathcal{F}^\pi$ has a unique fixed point $G^\pi_\sim$ in the compact state space of MVRL. This aggregator $\phi$ serve as mapping that transforms multi-view observations into a more compact space of summarizations $\mathcal{Z}$, defined as: $\phi : \mathcal{O}^1 \times \mathcal{O}^2 \times \cdots \times \mathcal{O}^k \to \mathcal{Z}$, which clusters inputs that are predicted to be similar under the learned bisimulation metric. Thus, the original multi-view RL problem can be approximated by solving a *summarized MDP* $\langle \mathcal{Z}, \mathcal{A}, \mathcal{P}, \Omega, \mathcal{R}, \gamma, p_0 \rangle$. Any RL algorithms can be applied to solve for the policy $\pi$ in this summarized space, and the learned policy can be evaluated in the original multi-view setting by selecting actions according to $\pi(\cdot | \phi(\vec{o}))$.

## 4.2 THEORETICAL ANALYSIS

In this section, we present a theoretical analysis: applying standard RL algorithms to a *summarized MDP*, which aggregates multi-view observations based on the behavioral similarity of their learned representations, can significantly improve sample complexity guarantees, provided that the learned representations incorporate bisimulation metrics.

**Lemma 1.** *Given a summarized MDP constructed by a learned aggregator $\phi : \mathcal{O}^1 \times \mathcal{O}^2 \times \cdots \times \mathcal{O}^k \to \mathcal{Z}$ that clusters multi-view observations in a $\epsilon$-neighborhood. The optimal value functions of original MDP and the summarized MDP are bounded as:*

$$|V^*(\vec{o}) - V^*(\phi(\vec{o}))| \leq \frac{2\epsilon}{(1 - \gamma)(1 - c)}. \tag{3}$$

The proof can be found in the Appendix A. This Lemma 1 serves to establish a bound on the difference between the optimal value functions of multi-view observations and their corresponding clusters in a simplified MDP, induced by a learned aggregator $\phi$. Specifically, it quantifies the impact of clustering errors and discrepancies in distance calculations on the value function, providing a controlled upper bound for these differences. The lemma highlights that by leveraging the learned aggregator, one can effectively reduce the complexity of the multi-view MDP's state space while maintaining a predictable level of accuracy in value function estimation.

## 5 LEARNING FUSED STATE REPRESENTATIONS FROM MULTI-VIEW OBSERVATIONS

As analyzed in the aforementioned Section 4, a critical component for achieving sample efficiency in RL algorithms is the aggregator $\phi : \mathcal{O}^1 \times \mathcal{O}^2 \times \cdots \times \mathcal{O}^k \to \mathbb{R}^d$, which is capable of learning task-relevant representations from multi-view observations. In this section, we describe the implementation details of the aggregator $\phi$. Specifically, our methods consists of two components: (a) **Self-Attention Fusion Module Combining Bisimulation Metrics**, which helps the aggregator capture task-relevant representations from multi-view observations; and (b) **Mask and Latent Resconstruction**, which is a an auxiliary objective of representation learning to promote cross-view state aggregation. The framework of our method is depicted in Figure 1.

### 5.1 SELF-ATTENTION FUSION MODULE COMBINING BISIMULATION METRICS

In multi-view RL, although observations from different views can provide the agent with diverse control information, they inadvertently increase the complexity of information extraction and aggregation. Prior studies have demonstrated that bisimulation metric serve as a useful form of state abstraction to capture task-representations from high-dimensional observation space in single-view RL task. In this paper, we found that bisimulation metric can also be applied to multi-view RL, resulting in a significant enhancement in performance. Specifically, our approach to bisimulation for multi-view representation learning consists of two main submodules: (a) **Convolutional Feature Embedding**, generating embeddings of the original high-dimensional multi-view observations; (b) **Self-Attention Fusion Module**, learning and integrating multi-view representations based on bisimulation metric.

**Convolutional Feature Embedding.** The feature encoding module uses a Convolutional Neural Network (CNN) encoder to encode single-view image observations into fixed-dimensional embeddings. Given a multi-view observations $\vec{O} = \{O^1, O^2, \ldots, O^k\}$, where $O^i \in \mathbb{R}^{H \times W \times C}$, the CNN encodes each image into a single-view representation $x^i$, where $x^i \in \mathbb{R}^d$.

**Self-Attention Fusion Module.** Similar to the `[class]` token used in BERT Devlin (2018) and ViT Dosovitskiy (2020), we prepend a learnable state fusion embedding $x^0 \in \mathbb{R}^d$ to the sequence of multi-view embedded representations. The state fusion representation $x^0$ is learned through self-attention mechanism and bisimulation metric, serving as the final fused representation of the multi-view observations, which is also used for training downstream RL algorithms. Additionally, position embeddings are added to the sequence of multi-view observation embeddings to retain view-specific information:

$$z_0 = [x^0, x^1, x^2, \ldots, x^k] + E_{pos}. \tag{4}$$

We utilize standard learnable 1D position embeddings. The embedded sequence is then fed into the Self-Attention Fusion Module. Specifically, the Self-Attentioni Fusion Module consists of $L$ attention layers. Each layer is composed of a Multi-Headed Self-Attention (MSA) layer, a layer normalization (LN)s, and Multi Layer Perceptron (MLP) blocks. The process can be described as follows:

$$z'_\ell = \text{MSA}(\text{LN}(z_{\ell-1})) + z_{\ell-1}, \quad \ell = 1 \dots L \tag{5}$$

$$z_\ell = \text{MLP}(\text{LN}(z'_\ell)) + z'_\ell. \quad \ell = 1 \dots L \tag{6}$$

The output after $L$ attention layers is $z_L = \{x_L^0, x_L^1, \dots, x_L^k\}$, where $x_L^0$ represents the final state fusion embedding. Therefore, we can define the fusion state of multi-view observations $\vec{o}$ aggregated by the aggregator $\phi$ as: $s = \phi(\vec{o})$. To capture the task-relevant representations from multi-view observations, bisimulation metirc learning is introduced in the process of state fusion. Consider bisimulation metric on policy $\pi$ in Equation 1, the measurement $G$, as in SimSR (Zang et al. (2022)), is defined using cosine distance, which has lower computational complexity compared to the Wasserstein distance and effectively prevents representation collapse.

In RL, we can view the critic in actor-critic algorithms such as SAC (Haarnoja et al. (2018)), as being composed of two function approximators $\psi$ and $\phi$, with parameters $\theta$ and $\omega$ respectively: $Q_{\theta,\omega} = \psi_\theta(\phi_\omega(\vec{o}))$. Here, $\psi_\theta$ serves as the value function approximator, while $\phi_\omega$ is the state aggregator, with the goal of aligning the distances between representations to match the cosine distance. Therefore, the parameterized representation distance $G_{\phi_\omega}$ can be defined as an approximant to the original observation distance $G^\pi$:

$$G^\pi(\vec{o}_i, \vec{o}_j) \approx G_{\phi_\omega}(\vec{o}_i, \vec{o}_j) := 1 - \cos_{\phi_\omega}(\vec{o}_i, \vec{o}_j) = 1 - \frac{\phi_\omega(\vec{o}_i)^T \cdot \phi_\omega(\vec{o}_j)}{\|\phi_\omega(\vec{o}_i)\| \cdot \|\phi_\omega(\vec{o}_j)\|}. \tag{7}$$

Based on Equation 2, the objective of state fusion with bisimulation metirc is:

$$\mathcal{L}_{fus} = \mathbb{E}_{(\vec{o}_i, r(\vec{o}_i, a), a, \vec{o}'_i), (\vec{o}_j, r(\vec{o}_j, a), a, \vec{o}'_j) \sim \mathcal{D}} \left( G_{\phi_\omega}(\vec{o}_i, \vec{o}_j) - \text{Target} \right)^2, \tag{8}$$

where Target $= \left| r^\pi_{\vec{o}_i} - r^\pi_{\vec{o}_j} \right| + \gamma G_{\phi_\omega}(s'_i, s'_j), s'_i \sim \hat{\mathcal{P}}\left(\cdot \left| \phi_\omega(\vec{o}'_i), a\right), s'_j \sim \hat{\mathcal{P}}\left(\cdot \left| \phi_\omega(\vec{o}'_j), a\right)\right.$. $\hat{\mathcal{P}}$ is latent state dynamics model. For detailed explanations on the training process of the latent state dynamics model and the reward scaling mechanism, please refer to the Appendix C.1 and C.2. $\mathcal{D}$ is the replay buffer. By incorporating bisimulation metrics during state aggregation, our model is able to focus on the causal features that directly influence rewards, effectively integrating information from multi views. As a result, the learned representations are both compact and highly task-relevant.

## 5.2 MASK AND LATENT RECONSTRUCTION

To learn more compact and task-relevant representations from multi-view observations, we employed a Mask-based Latent Reconstruction strategy in addition to bisimulation metric learning. In visual RL tasks, previous works (Yu et al. (2022a), Wei et al. (2022)) have shown that the significant spatio-temporal redundancy can be eliminated by mask-based reconstruction methods. Consequently, we reconstruct spatially masked pixels in the latent space by leveraging potential correlations between multiple views. Compared to reconstruction in the original pixel space, reconstructing the inferred state representations from the unmasked frames preserves essential state control information while reducing unnecessary spatial redundancy.

Specifically, we randomly masked a portion of the original multi-view image observations $\{O^1, O^2, \dots, O^k\}$. The masked observation sequences $\{\tilde{O}^1, \tilde{O}^2, \dots, \tilde{O}^k\}$ is then processed through the CNN Encoder and the Self-Attention Fusion Module, resulting in a set of masked state embeddings $\{\tilde{x}_L^0, \tilde{x}_L^1, \dots, \tilde{x}_L^k\}$. Motivated by the success of SimSiam Chen & He (2021) in self-supervised learning, we use an asymmetric architecture for calculating the distance between the reconstructed latent states and the target states. The masked state embeddings are passed through a prediction head to get the final reconstructed/predicted state $\{\hat{x}_L^0, \hat{x}_L^1, \dots, \hat{x}_L^k\}$. We construct the reconstruction loss using cosine similarity, ensuring that the final predicted result closely approximates its corresponding target. The final objective function of Mask and Latent Reconstruction can be formulated as:

$$\mathcal{L}_{res} = 1 - \frac{1}{k+1} \sum_{i=0}^{k} \frac{(\hat{x}_L^i)^T \cdot x_L^i}{\|\hat{x}_L^i\| \cdot \|x_L^i\|}. \tag{9}$$

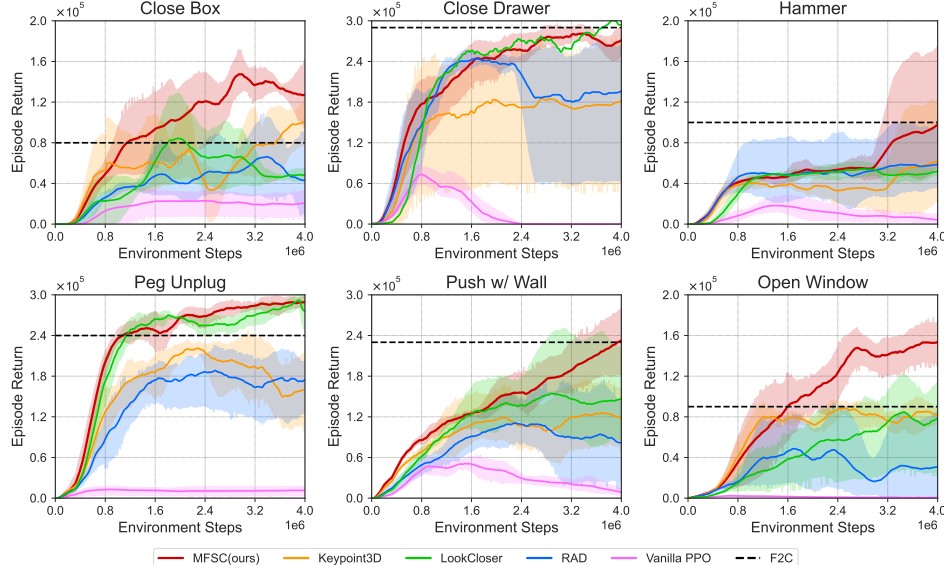

Figure 2: Performance comparison on six robotic arm manipulation tasks from Meta-World. All curves of show the mean and its 95% Confidence Intervals (CIs) of performance across 4 independent seeds. The black dashed line represents the final convergence result of F2C in Meta-World.

The Mask-based Latent Reconstruction serves as an auxiliary task, and is optimized together with multi-view state fusion module. Thus, the overall loss function is:

$$\mathcal{L}_{total} = \mathcal{L}_{fus} + \mathcal{L}_{res}. \tag{10}$$

## 6 EXPERIMENT

Through our experiments, we aim to investigate the following questions: (1) How does MFSC perform in multi-view observation learning compared to existing methods? (2) Can we learn effective state representations for planning under multi-view observations in high-dimensional control tasks with insufficient guiding information? (3) To what extent can MFSC handle tasks with missing views? Lastly, we present visualization and ablation studies to demonstrate the model's attention to different views and the effectiveness of its components.

### 6.1 SETUP

**Experimental Setup** We evaluated our method across multiple 3D control tasks using pixel observations from three cameras. We selected a set of 3D manipulation environments (Yu et al. (2020)) and a high degree-of-freedom 3D locomotion environment (Coumans & Bai (2022)). These environments were originally designed for state-based reinforcement learning (RL), posing significant challenges for pixel-based RL. Additionally, we conducted tests involving missing guiding colors and views, as well as related visualization experiments.

**Baselines** We compared MFSC against several baseline methods. All baselines, including MFSC, were implemented using PPO (Schulman et al. (2017)). The baselines include: (1) Keypoint3D (Chen et al. (2021)), which uses keypoint detection to reconstruct views based on learned keypoints; (2) LookCloser (Jangir et al. (2022)), which applies cross-attention between pairs of views to integrate multi-view information; (3) Fuse2Control (F2C) (Hwang et al. (2023)), which employs an information-theoretic approach to learn a state space model and extract information independently from each view. Additionally, for common RL algorithms, we stack images from all three views to form the observation: (4) RAD (Laskin et al. (2020b)), which achieves high sample efficiency through data augmentation; (5) Vanilla PPO (Schulman et al. (2017)), the original PPO (Schulman et al. (2017)) algorithm, using a CNN architecture to process image observations.

### 6.2 CONTROL WITH MULTI-VIEW OBSERVATIONS

For fair comparison, we adopt the same experimental setup as (Chen et al. (2021)). We employed six robotic arm manipulation tasks from the Meta-World benchmark, each featuring 50 random con-

figurations. Details regarding the specific task settings and our treatment of the reward function can be found in the Appendix C. As shown in Figure 2, our method consistently outperforms state-of-the-art techniques across all six environments, exhibiting significantly higher sample efficiency and demonstrating more stable performance compared to other approaches. Vanilla-PPO, in particular, showed almost no signs of learning in vision-based environments, indicating the difficulty of extracting meaningful state representations without auxiliary tasks. RAD generally performs well in simpler tasks; however, it struggles to learn effective fused representations in tasks such as *'Open Window'* and *'Close Box'* where task completion does not rely on a specific view. Keypoint3D demonstrated competitive performance in certain tasks, especially in *'Close Box'*, but overall, its training efficiency and final performance were suboptimal, requiring additional view-specific information. The cross-attention encoder, also based on a Transformer architecture, proved to be effective as well. LookCloser performs well in some tasks (*'Close Drawer'* and *'Peg Unplug'*), but overall performance is not as good as MFSC. F2C, as a leading MVRL method, and MFSC both demonstrated strong competitiveness in extracting control-relevant state representations, underscoring the importance of learning efficient representations from high-dimensional multi-view observations.

### 6.3 Scaling to High-Dimensional Control

To further evaluate the performance of our method in high-dimensional control tasks, we conducted experiments in the highly dynamic 3D locomotion environment of Pybullet's Ant. This environment requires controlling multiple movable joints and involves complex dynamics, necessitating a detailed understanding of the movable joints and components from multi-view observations. Given the temporal reasoning required in this locomotion task, we utilized a frame stack of 2. Additionally, in the original Ant environment, Pybullet assigns different colors to adjacent limbs to aid the algorithm in capturing key information related to the Ant's movements. To further validate whether our algorithm can still capture task-relevant information in the absence of explicit visual cues, we conducted benchmark tests in a colorless version of the Ant environment, following the approach of Keypoint3D.

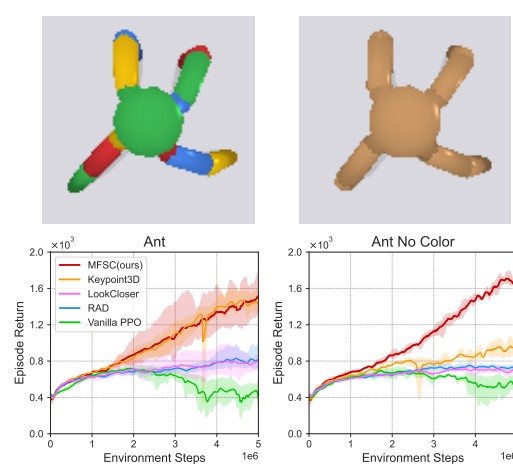

Figure 3: Ant performance in high-dimensional control tasks.

As shown in Figure 3, our method performs on par with the Keypoint approach in the colored Ant environment and significantly outperforms all baseline methods in the colorless version. During training, our algorithm exhibited stable and consistent performance improvement, effectively avoiding local minima. Even in the colorless version, where key visual cues are absent, our method maintained strong performance, demonstrating its ability to effectively capture and aggregate task-relevant information from multi-view observations. In contrast, Vanilla PPO and RAD exhibited limitations in extracting relevant information. Methods based on contrastive learning and reconstruction tend to focus excessively on local pixel changes, failing to capture fine-grained, task-critical information. This robustness underscores the broad applicability of our approach, ensuring reliable performance even in environments with limited visual textures, particularly in high-dimensional, low-texture settings.

### 6.4 Robustness against missing views

While missing-view tasks inevitably result in the loss of some crucial state-related information, we systematically evaluated the performance of MFSC under missing-view conditions on three tasks from the Meta-World benchmark. During training, we explicitly introduced a mask token for the missing views, which was input alongside the representations from other views to maintain cross-view information exchange and fusion. In the testing phase, we employed a strategy of randomly omitting one of the view frames to simulate the real-world scenarios where view information may be incomplete. Figure 4 summarizes the performance comparison between MFSC under missing-view

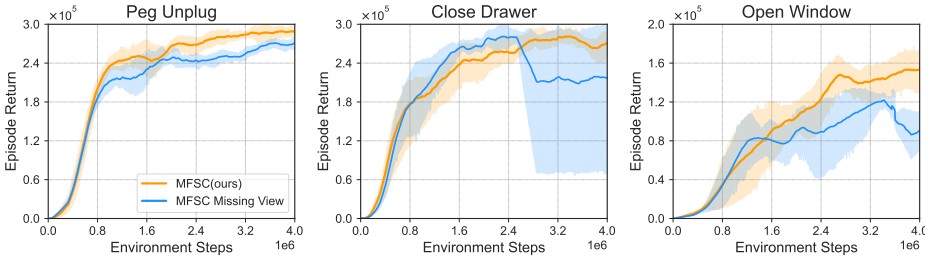

Figure 4: Performance comparison of MFSC under full-view and missing-view conditions

and full-view conditions. The results indicate that, although missing views may impact certain task-specific details—such as the precise representation of object position or robotic arm posture—the performance degradation of MFSC is relatively limited across the tested tasks. Our method demonstrates significant robustness to missing views. Even with partial view information, MFSC is able to leverage cross-view consistency to learn effective task-relevant representations. This robustness is largely attributed to the effective fusion of multi-view information during training, particularly through the introduction of the mask token mechanism. This allows our model to maintain high performance even in scenarios with incomplete information.

## 6.5 ABLATION STUDY AND ANALYSIS

Figure 5 illustrates the cumulative returns of the algorithm across two benchmarks, Meta-World and Pybullet, comparing three variations: MFSC (the proposed method), MFSC without bisimulation constraints ('MFSC w/o bis'), and MFSC without Mask and Latent Reconstruction ('MFSC w/o res'). The curves represent the mean performance, with the shaded areas indicating the variance across trials. MFSC (red line), as the complete method,

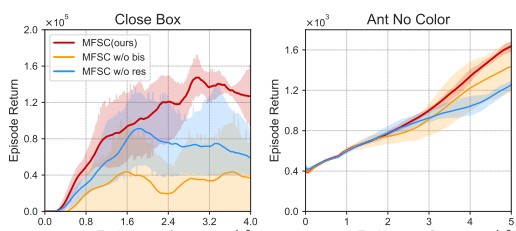

Figure 5: Performance of ablation study.

achieves the highest cumulative returns throughout the process. As the number of environment steps increases, the model's performance steadily improves. The relatively small variance suggests that MFSC excels not only in learning optimal control policies but also demonstrates high robustness. In contrast, removing the bisimulation constraint in MFSC significantly degrades performance. This ablation study highlights the importance of the bisimulation component in MFSC, as its absence results in earlier performance plateauing and notably poorer returns. Additionally, the larger variance indicates that the strategy without bisimulation is not only suboptimal but also less consistent. 'MFSC w/o res' (blue line) performs better than 'MFSC w/o bis' but still falls short of the full MFSC method. Although its variance is slightly higher than MFSC, it exhibits much less fluctuation compared to MFSC without bisimulation, indirectly emphasizing the significance of learning cross-view information.

## 6.6 WHAT EXACTLY IS MFSC FOCUSING ON WITHIN AND ACROSS VIEWS?

We employ Grad-CAM (Selvaraju et al. (2017)) to visualize the learned representations of MFSC. Our primary objective is to investigate whether MFSC can address the two challenges mentioned at the outset: 1) whether MFSC can effectively capture task-relevant information in multi-view observations that contain higher data dimensions and more redundant information; and 2) whether MFSC can facilitate an informative aggregation of representations across various views.

For *Challenge 1*, we conduct a separate gradient analysis for each view. Grad-CAM heatmaps are generated based on gradients computed from the bisimulation loss. Subsequently, we apply min-max normalization to the heatmaps for each individual view. As shown in the visualizations (middle column of each frame), MFSC consistently focuses on task-relevant features—such as the target position, the robotic arm, or the ant's legs—while paying less attention to elements less relevant to control, such as the window edges or the ant's body. From this analysis, we infer that MFSC is capable of successfully identifying and extracting task-relevant information from each view.

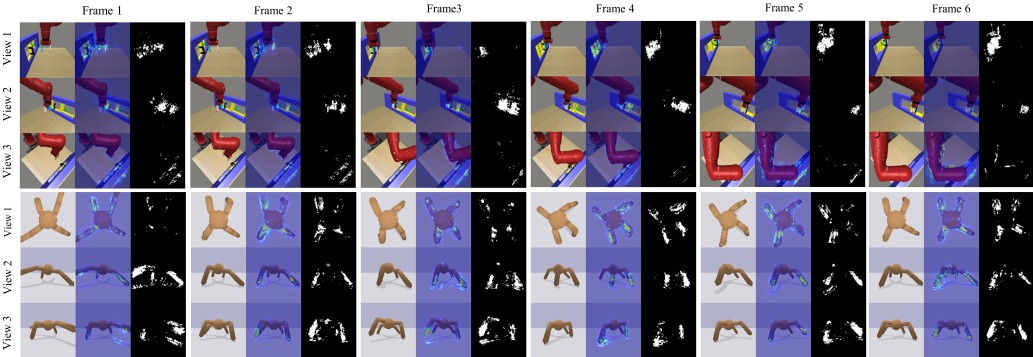

Figure 6: Visualization of multiple views for task-specific aggregation.

For *Challenge 2*, we perform a joint analysis of the three views. We select the pixels with the highest gradient values across the three views, marking them as 1 (white), while marking the remaining pixels as 0 (black). In the *'Open Window'* task (first row), at the beginning of the task, when the goal is to move the robotic arm to the correct position, all three views contain significant information, and the model allocates its attention accordingly across the views. However, when the task shifts to opening the window, occlusion occurs in the third view. Despite the larger spatial presence of the arm in the third view (top view), the model shifts its attention to the first and second views, which contain more task-relevant information. In the Ant-no-color task, the information from all three views is relatively important, as indicated by the relatively uniform distribution of top gradient pixels across the three views. This suggests that MFSC allocates its attention more evenly across the views in the ant task.

The visualization above demonstrates that MFSC can effectively aggregate representations from multiple views, allowing task-relevant features to be extracted from different perspectives. This aggregation enhances the performance of downstream reinforcement learning tasks by providing a more comprehensive and fused understanding of the environment. MFSC's ability to integrate and align information from diverse observational inputs enables more efficient policy learning and decision-making in complex control scenarios.

## 7 DISCUSSION

**Limitations and Future Work.** In addressing missing views, we applied masking techniques akin to those used in natural language processing. However, the absence of critical views in reinforcement learning tasks can have a significant impact. For certain views containing key information, even with inference from other observations, accurately reconstructing the true environmental state remains challenging. This is because the information across multiple views may not be entirely complementary, particularly in situations involving complex state transitions or occlusions. Under such conditions, the current method may not provide sufficient robustness. To tackle the issue of missing views, future research could explore incorporating state-space models to better capture temporal dependencies, enabling more accurate state estimation in the absence of certain views. Additionally, expanding the model's capability to process multimodal inputs is a promising direction. For instance, integrating real-world sensor data with image observations and leveraging multimodal information could enhance the RL agent's perception and decision-making capabilities in complex environments.

**Conclusion.** We propose a novel framework, Multi-view Fusion State for Control (MFSC), to address the challenge of learning task-relevant representations in Multi-View Reinforcement Learning (MVRL). MFSC combines self-attention mechanisms with bisimulation metric learning to fuse multi-view observations while maintaining task relevance. Additionally, MFSC introduces a mask-based latent space reconstruction auxiliary task to enhance the model's ability to capture cross-view information and improve the learned representations. Experimental results on Meta-World and Py-bullet benchmarks demonstrate that MFSC effectively aggregates task-relevant details and shows robustness in scenarios with missing views. Finally, visualization analyses confirm MFSC's capability to capture task-relevant information and dynamically fuse multiple views.

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

## A  PROOFS

**Theorem 1.** Given a summarized MDP constructed by a learned aggregator $\phi : \mathcal{O}^1 \times \mathcal{O}^2 \times \cdots \times \mathcal{O}^k \to \mathcal{Z}$ that clusters multi-view observations in a $\epsilon$-neighborhood. The optimal value functions of original MDP and the summarized MDP are bounded as

$$|V^*(\vec{o}) - V^*(\phi(\vec{o}))| \leq \frac{2\epsilon}{(1-\gamma)(1-c)}. \tag{11}$$

*Proof.* The proof follows straightforwardly from DBC Zhang et al. (2021). From Theorem 5.1 in Ferns et al. (2004) we have:

$$(1-c)|V^*(\vec{o}) - V^*(\phi(\vec{o}))| \leq g(\vec{o}, \tilde{d}) + \frac{\gamma}{1-\gamma} \max_{u \in \mathcal{O}^1 \times \mathcal{O}^2 \times \cdots \times \mathcal{O}^k} g(u, \tilde{d}), \tag{12}$$

where $g$ is the average distance between a multi-view observation and all other multi-view observations in its equivalence class under the bisimulation metric $\tilde{d}$. By specifying a $\epsilon$-neighborhood for each cluster of multi-view observations, we can replace $g$:

$$(1-c)|V^*(\vec{o}) - V^*(\phi(\vec{o}))| \leq 2\epsilon + \frac{\gamma}{1-\gamma}2\epsilon$$

$$|V^*(\vec{o}) - V^*(\phi(\vec{o}))| \leq \frac{1}{1-c}\left(2\epsilon + \frac{\gamma}{1-\gamma}2\epsilon\right)$$

$$= \frac{2\epsilon}{(1-\gamma)(1-c)}.$$

As $\epsilon \to 0$, the optimal value function of the aggregated MDP converges to the original value function. By defining a learning error for $\phi$, $\mathcal{L} := \sup_{\vec{o}_i, \vec{o}_j \in \mathcal{O}^1 \times \mathcal{O}^2 \times \cdots \times \mathcal{O}^k} \left| ||\phi(\vec{o}_i) - \phi(\vec{o}_j)||_1 - \tilde{d}(\vec{o}_i, \vec{o}_j) \right|$, we can also update the bound in Lemma 1 to incorporate $\mathcal{L}$ : $|V^*(\vec{o}) - V^*(\phi(\vec{o}))| \leq \frac{2\epsilon + 2\mathcal{L}}{(1-\gamma)(1-c)}$.

## B  EXTENDED RELATED WORK

For the sake of brevity, we previously provided a high-level overview of the related work on state representation learning in RL. We now offer a more detailed discussion. Well-constructed state representations enable agents to better comprehend and adapt to complex environments, thereby improving task performance and decision-making efficiency. For instance, methods such as CURL (Laskin et al. (2020a)) and DrQ (Kostrikov et al. (2020), Yarats et al. (2021)) leverage data augmentation techniques like cropping and color jittering to enhance model generalization. However, their performance is highly dependent on the specific augmentations applied, leading to variability in results. Masking-based approaches (Seo et al. (2023b), Yu et al. (2022b), Seo et al. (2023a), Liu et al. (2022)) selectively obscure parts of the input to mitigate redundant information and improve training efficiency. While these methods show promise in filtering out irrelevant data, they carry the risk of unintentionally discarding task-critical information, potentially affecting overall agent performance. Bisimulation-based strategies (Zhang et al. (2021), Zang et al. (2022)) focus on constructing reward-sensitive state representations to ensure that states with similar values are close in the representation space, promoting sample efficiency and consistent decision-making. Another line of research explores causal relationships between state representations and control (Wang et al. (2022), Lamb et al. (2022), Efroni et al. (2021), Efroni et al. (2022), Fu et al. (2021)). By analyzing the causal links between states and actions, these methods aim to improve agents' understanding and control of the environment, further optimizing RL performance.

## C  EXPERIMENTAL DETAILS

Table 1 provide detailed information regarding the experimental setup and hyperparameter configurations. Our model architecture adheres to the PPO-based design proposed by Chen et al. (2021). In the Metaworld environment, we utilize a representation size of 128, following the Keypoint3D framework outlined by Chen et al. (2021). All networks in both the policy and representation models are optimized using the Adam optimizer (Kingma (2014)), ensuring consistent performance across various environments.

Table 1: MFSC's hyperparameters, based on PPO.

| Hyperparameter | Meta-World | Ant |
|---|---|---|
| **General** | | |
| PPO batch size | 6400 | 16000 |
| Rollout buffer size | 100000 | 100000 |
| Epochs per update | 8 | 10 |
| Gamma | 0.99 | 0.99 |
| GAE lambda | 0.95 | 0.95 |
| Clip range ($\epsilon$) | 0.2 | 0.2 |
| Entropy coefficient | 0.0 | 0.0 |
| Value function coefficient | 0.5 | 0.5 |
| Gradient clip | 0.5 | 0.5 |
| Target KL | 0.12 | 0.12 |
| Policy learning rate | $2 \cdot 10^{-4}$ | $2 \cdot 10^{-4}$ |
| **MFSC** | | |
| State representation dimension | 128 | 128 |
| Weight of fusion loss ($\lambda_{fus}$) | 1.0 | 1.0 |
| Weight of reconstruction loss ($\lambda_{res}$) | 1.0 | 1.0 |
| Number of dynamics models | 5 | 5 |
| Mask ratio | 0.8 | 0.8 |
| Cube spatial size | $12 \times 12$ | $12 \times 12$ |
| Cube depth | 3 | 3 |
| Self-attention fusion module depth | 2 | 2 |

## C.1 LATENT STATE DYNAMICS MODELING

Following our approach, we develop an ensemble version of deterministic dynamics models $\{\hat{\mathcal{P}}_k(\cdot|\phi_\omega(x), a)\}_{k=1}^K$. Unlike probabilistic dynamics models, our transition models are deterministic and their outputs are consistent with the encoder's output, both of which are subjected to $l_2$-normalization. Instead of using a probabilistic transition, we calculate the distance using cosine similarity. Specifically, at the training step, we update the parameters of the dynamics models based on the cosine similarity loss function:

$$\mathcal{L}_{dyn} = \frac{1}{K} \sum_{k=1}^K \left[ 1 - \frac{\hat{\mathcal{P}}_k(\cdot|\phi_\omega(\vec{o}), a) \cdot \phi_\omega(\vec{o'})}{\|\hat{\mathcal{P}}_k(\cdot|\phi_\omega(\vec{o}), a)\| \|\phi_\omega(\vec{o'})\|} \right] \tag{13}$$

where $i \in \{1, 2, \ldots, K\}$. Since the deterministic models share the same gradient but are initialized randomly, they may still acquire different parameters after training. This ensemble model allows us to estimate the latent dynamics of the environment effectively while ensuring the output remains consistent across the encoder and dynamics model. At the inference step, we randomly sample one of the $K$ deterministic dynamics models to compute the transition to the next latent state $s'$.

## C.2 REWARD NORMALIZATION

Reward normalization is a crucial component of our representation learning approach, as it directly relies on the reward function to guide feature extraction and learning. In the experimental tasks, the rewards used for representation learning are consistent with those used in policy learning. Following Keypoint3D (Chen et al. (2021)), to ensure stable learning dynamics, we apply a moving average normalization method to dynamically normalize the reward values. This method calculates the moving average of historical rewards and adjusts the rewards to have a mean of 0 and a standard deviation of 1. This normalization process helps mitigate fluctuations in reward values caused by variations in task difficulty, environmental changes, or exploration strategies, enabling the model to more effectively learn meaningful representations from stable reward signals. Additionally, since the scale of rewards influences the bisimulation metric and the upper bound of value function errors, we adopt reward scaling to avoid feature collapse and reduce bisimulation measurement errors.

Following the work of Zang et al. (2023), we scale the normalized rewards. Rather than using the conventional settings of $c_r = 1$ and $c_k = \gamma$, we apply $c_r = 1 - \gamma$ and $c_k = \gamma$ to scale the normalized rewards effectively.

### C.3 META-WORLD

To evaluate whether our model can accelerate policy optimization when jointly trained with the policy, we conducted six complex robotic arm manipulation tasks in the Metaworld environment (Yu et al. (2020)). Each task involves 50 randomized configurations, such as the initial pose of the robot, object locations, and target positions. For each task, we utilized three third-person cameras from different angles to observe the robot arm and relevant objects. Since the state of the gripper at the end of the robotic arm may not be clearly visible from any of the three camera angles, following the settings of Chen et al. (2021) and Hwang et al. (2023), an indicator was introduced in the Metaworld tasks to signify whether the gripper is open or closed. This indicator is concatenated with the learned latent state and fed into the policy network. Due to experimental variations, we adopted the results reported in the F2C paper for comparison.

### C.4 PYBULLET-ANT

The PyBullet Ant (Coumans & Bai (2022)) task is designed to simulate the motion control of a quadruped robot in a highly dynamic 3D-locomotion environment. The objective of this task is to control the joints of the robot's legs, enabling it to learn how to balance and move as quickly and stably as possible. The Ant robot has a highly dimensional state and action space, which includes physical quantities such as joint angles, angular velocities, and linear velocities. The robot's movement is generated by controlling the torque or force applied to its joints, making a fine-grained understanding of the movable joints and parts essential. As locomotion environments require temporal reasoning, we use a frame stack of 2. The reward function in this task is typically based on the robot's forward velocity, while accounting for control costs (energy consumption), to incentivize efficient movement. Due to the complexity of the environment and the high-dimensional action space, the Ant task provides a significant challenge for training and testing reinforcement learning algorithms.

## D ALGORITHM

Our traning algorithm is shown in Algorithm 1.

---

**Algorithm 1** MFSC: Multi-view Fusion State for Control

---

**Input**: $N_{\text{Repeat}}$ # of iterations to repeat entire processes.
       $B$ batch size, $T$ rollout length.

1: **for** iter = 1 to $N_{\text{Repeat}}$ **do**
2:     **Initialize** $\mathcal{B}_{rollout}$.
3:     **for** $b = 1$ to $B$ **do**
4:         Run policy $\pi_{\theta_{\text{old}}}$ to collect $(\vec{o}, a, r, \vec{o}')_{1:T}$
5:         $\mathcal{B}_{rollout} \leftarrow \mathcal{B}_{rollout} \cup (\vec{o}, a, r, \vec{o}')_{1:T}$
6:     **end for**
7:     Estimate advantage values $\hat{A}_{1:T,1:N}$ on $\mathcal{B}_{rollout}$
8:     **for** $t = 1$ to $T$ **do**
9:         Sample $(\vec{o}, a, r, \vec{o}') \sim \mathcal{B}_{rollout}$
10:       Cube masking the multi-view observation $\vec{o}$
11:       Calculate $\mathcal{L}_{rec}$ according to Eq.8
12:       Calculate $\mathcal{L}_{fus}$ according to Eq.9
13:       Calculate $\mathcal{L}_{dyn}$ according to Eq.13
14:       Optimize $\mathcal{L}_{policy} + \mathcal{L}_{rec} + \mathcal{L}_{fus} + \mathcal{L}_{dyn}$ throughout $\mathcal{B}_{rollout}$
15:     **end for**
16:     $\pi_{\text{old}} \leftarrow \pi$
17: **end for**

---

