# OpenReview forum: "Learning Fused State Representations for Control from Multi-View Observations"
_ICLR.cc/2025/Conference — ICLR 2025 Conference Withdrawn Submission_

### Official Review · Reviewer_HetD · 2024-10-28

**Soundness:** 3
**Presentation:** 3
**Contribution:** 2
**Rating:** 3
**Confidence:** 3

**Summary:**

The paper introduces a novel approach named Multi-view Fusion State for Control(MFSC)，which ingrates a self-attention mechanism with bisimulation metric learning to fuse task-relevant representation from multi-view observation. Additionally, the paper also incorporated a mask-based latent reconstruction auxiliary task to learn cross-view information in order to foster more compact fused presentation. In this paper, two major problems were solved : First is Higher data dimensions and more redundant information , and Informative aggregation of representation from various views.

**Strengths:**

1.	Clear statements and good structure. The paper is well-structured, and viewpoints was stated logically. The introduction provides a good overview of the challenges in the multi-view representation learning task and approach to address them relatively.  Also illustrate provided along with methods made it easy and vivid.
2.	Sufficient and solid proof in major conclusions. Problems were clearly defined and followed by mathematical formulations with clear explanation and ended with a solution with validate experiments.
3.	Comprehensive experiment and supportive solution ,also contributions made by this method were shown vividly and clearly through several comparative illustrate shown in the part of Experiments.
4.	Reproductive experiment with project code and data shared.  Experiments  result can be verified personally by readers with resources provided in this paper.

**Weaknesses:**

•	A few formula faults are discovered in the paper.
•	Evaluation Metrics: The evaluation metrics used in the experiments could be more comprehensive. Currently, the focus appears to be on task performance, but including metrics that assess representation quality (e.g., reconstruction loss) would provide a fuller picture of the model’s effectiveness.
•	Generalization to Other Tasks: The experiments are primarily conducted on Meta-World. To evaluate the generality of the approach, the authors should consider applying MFSC to other control tasks or environments. This would help demonstrate the versatility and broader applicability of the proposed method.
•	Limitations Discussion: The paper should include a dedicated section discussing the limitations of the proposed method. Identifying potential weaknesses and suggesting avenues for future work would add depth to the contribution.

**Questions:**

Overall, while the MFSC architecture presents a promising direction for multi-view reinforcement learning, addressing the outlined weaknesses and incorporating the suggested improvements will significantly enhance the paper's clarity, depth, and impact in the field.

---

> ### Comment · Reviewer_HetD · 2024-11-26
> **To authors**
>
> I would like to decrease my score to 3.

---

### Official Review · Reviewer_Tb2V · 2024-11-04

**Soundness:** 3
**Presentation:** 3
**Contribution:** 3
**Rating:** 5
**Confidence:** 3

**Summary:**

The paper presents a novel architecture called Multi-view Fusion State for Control (MFSC), designed to learn compact and task-relevant representations from multi-view observations in reinforcement learning (RL). This approach integrates a self-attention fusion module with bisimulation metric learning to aggregate information from different views, while also using a mask-based latent reconstruction auxiliary task to promote cross-view information aggregation.  Experiments conducted on Meta-World and Pybullet demonstrate the superiority of MFSC over other methods.

**Strengths:**

1.	The paper addresses the challenging and significant problem of learning task-relevant fused state representations from multi-view observations, which is a crucial aspect of multi-view reinforcement learning.
2.	The integration of a mask-based latent reconstruction task enhances the model’s ability to learn cross-view information. The proposed approach, combining self-attention and bisimulation metrics, offers an effective solution.
3.	This paper demonstrates the effectiveness of MFSC across multiple challenging benchmarks, including robotic manipulation tasks in Meta-World and control tasks in Pybullet.

**Weaknesses:**

1.	This paper does not include comparisons with approaches tailed for visual RL, such as [1-2], particularly multi-view visual RL method like [3]. Evaluating MFSC against such baselines would provide a more accurate assessment of its effectiveness and novelty.
2.	How does the computational complexity of MFSC compare to baseline approaches in terms of training time, inference time, and resource requirements?
3.	This paper does not provide sensitivity analyses of MFSC with respect to different hyperparameters, such as the weight of fusion loss and the weight of reconstruction loss.
References
[1] Hafner et al. Mastering diverse domains through world models. arXiv preprint   arXiv:2301.04104, 2023.
[2] Seo et al. Masked world models for visual control. CORL, 2023.
[3] Seo et al. Multi-view masked world models for visual robotic manipulation. ICML, 2023.

**Questions:**

Please see weakness section.

---

### Official Review · Reviewer_1MQc · 2024-11-04

**Soundness:** 3
**Presentation:** 3
**Contribution:** 2
**Rating:** 5
**Confidence:** 4

**Summary:**

This paper proposes the Multi-view Fusion State for Control (MFSC), which integrates a self-attention mechanism and bisimulation metric learning to fuse task-relevant representations from multi-view observations, and incorporates a mask-based latent reconstruction auxiliary task to obtain more compact fused representations and handle missing views.

**Strengths:**

1. The writing is relatively clear.

2. The performance of the proposed method is validated on Meta-World and Pybullet benchmarks.

**Weaknesses:**

1. The author incorporates bisimulation principles by integrating reward signals and dynamic differences into the fused state representation to capture task-relevant details. As I am aware, [1] also acquires representations for control with bisimulation metrics. Additionally, the author employed a Mask-based Latent Reconstruction strategy, which is analogous to that in [2]. Does this similarity suggest a deficiency in significant innovation or does the author offer additional components or enhancements that differentiate it from the existing strategies in [1] and [2]? Furthermore, it is essential to determine whether appropriate credit and comparison with the prior works in [1] and [2] have been adequately accounted for.

[1] Learning invariant representations for reinforcement learning without reconstruction.

[2] Mask-based Latent Reconstruction for reinforcement learning。

3. Missing many recent visual RL baselines: the baselines used in the paper are all old methods and a large body of the recent methods developed on visual reinforcement learning are ignored [1][2].

[1] TACO: Temporal Latent Action-Driven Contrastive Loss for Visual Reinforcement Learning.

[2] Mastering Diverse Domains through World Models.

4. Whether this method is only useful for robot control tasks needs to be further verified on more types of environments, such as Carla, atari, etc.

5.  The paper lacks sufficient ablation experiments. The author only ablated MFSC without bisimulation constraints ('MFSC w/o bis') and MFSC without Mask and Latent Reconstruction ('MFSC w/o res'), but not more detailed parts like the Self-Attention Fusion Module.

6. The author claims that MFSC can be seamlessly integrated into any existing downstream reinforcement learning framework to enhance the agent's understanding of the environment. However, there are no relevant experiments to verify this claim.

**Questions:**

Please see the weaknesses.

---

### Official Review · Reviewer_56P8 · 2024-11-05

**Soundness:** 3
**Presentation:** 3
**Contribution:** 2
**Rating:** 5
**Confidence:** 4

**Summary:**

This paper proposes a method that combines a bisimulation-based approach with masked representation learning for multi-view reinforcement learning. The core idea is that to enable task-relevant multi-view fusion, it is essential to align the integration process closely with the specific objectives of the task. In other words, when fusing information from multiple views, the task’s specific goals (Equation 8) must be considered. The authors have evaluated their method on two visual control environments, including Meta-World and PyBullet, demonstrating significant performance improvements over baseline methods.

**Strengths:**

- The paper is clearly written and easy to understand.
- The proposed method that integrates bisimulation metric learning into the fusion process of multi-view states is reasonable.
- The authors have provided extensive experimental results, covering various visual RL environments, to validate the effectiveness of the method. The paper also includes experiments with missing views as well as additional visualizations to interpret the effectiveness of the method.

**Weaknesses:**

My main concerns involve the novelty of the method and the completeness of experimental comparisons:

- The primary limitation lies in the method's novelty. Although the authors present two core challenges of multi-view RL in the introduction, these challenges have already been extensively explored in prior research. While incorporating bisimulation metrics into state aggregation is reasonable, bisimulation-based methods are also well-covered in existing RL literature, making this combination feel more like a natural choice than a groundbreaking innovation.
- Although the authors conducted extensive experiments and validated the effectiveness of their approach against various existing multi-view RL methods, there are still two main gaps. First, there is no experimental verification of whether the method remains superior to baseline models in cases with missing views (even with a single view). Second, Seo et al. (2023) proposed the masked world model, which performs well on multi-view RL tasks and has methodological similarities to the approach in this paper. A direct comparison with Seo et al.'s work would provide stronger support for the effectiveness of this method.

**Questions:**

I recommend the authors systematically compare the similarities and differences between their method and Seo et al.'s masked multi-view RL approach within the main text.

---

### Author Response · Authors · 2024-12-02
**General Response to Reviewers and AC**

We thank all reviewers for their thoughtful comments. We would like to sincerely thank Reviewer 56P8, Reviewer 1MQc, and Reviewer HetD for your positive feedback on the clarity of our writing and the reasonableness of our proposed method. We also appreciate Reviewer Tb2V for highlighting the significance of the problem we address and for your kind words regarding the integration of mask-based latent reconstruction and the use of bisimulation metrics.

Furthermore, we would like to take this opportunity to elaborate on the contributions of our method and provide a comprehensive comparison with other related works.

1. Contribution of Our Method

Bisimulation has garnered significant attention as a method for learning robust representations in reinforcement learning. However, in the domain of multi-view fusion, the integration of bisimulation to learn fused multi-view state representations remains unexplored. To the best of our knowledge, this work pioneers the integration of multi-view state representation fusion with bisimulation metrics. Our method leverages self-attention mechanism and utilizes the output from the ViT architecture as the fused representation. By incorporating bisimulation metric learning into the representation fusion process, our approach dynamically extracts task-relevant features from each view and combines them based on their relevance. We believe our method offers a unified framework that addresses two critical challenges in multi-view representation learning: effective task-relevant feature extraction and dynamic information integration. This work provides new insights for progress in multi-view learning in the context of reinforcement learning.

2. Comparison with Other Related Works

Comparison with MV-MWM: In terms of learning objectives, MFSC uses bisimulation metric learning to extract task-relevant fused representations from multi-view observations, while MV-MWM employs a mask reconstruction task as an auxiliary objective. In terms of the masking strategy, MFSC reconstructs in latent space, avoiding the reconstruction of task-irrelevant details. In contrast, MV-MWM requires an additional decoder for pixel-level reconstruction to fully reconstruct all details from raw observations. Another notable distinction is that MV-MWM introduces expert data during the behavioral learning phase to guide policy optimization.

Comparison with DBC: First, we would like to clarify that DBC is not inherently a multi-view fusion method but rather a representation learning algorithm designed to enhance robustness. In contrast, our approach integrates self-attention mechanisms with bisimulation to effectively extract task-relevant fused representations from multi-view observations, addressing the traditional challenges associated with multi-view learning.

Comparison with MLR: To further enhance model learning capacity and reduce spatiotemporal redundancy, we employ a mask-based latent reconstruction strategy integrated to derive compact representations. Unlike MLR, MFSC incorporates a self-attention module within the encoder and employs a fusion mechanism to effectively learn fused state representations, which directly benefit downstream reinforcement learning tasks. In contrast, MLR leverages self-attention solely within the decoder and relies on an auxiliary loss term to guide convolutional neural networks (CNNs) in capturing temporal dependencies in sequences.

During the rebuttal phase, we conducted additional experiments as suggested by the reviewers, including comparisons with other baseline algorithms, and additional analyses on parameter sensitivity, representation quality (reconstruction loss and bisimilarity), training time, and inference time. We also verified the performance of our approach on Carla. Besides, we also tried to include a more comprehensive comparison with methods like MV-MWM and TACO. However, reproducing the MV-MWM method is not feasible because MV-MWM conducted their experiments on the RLbench benchmark, which necessitates unique designs and additionally used expert demonstration data. In the case of TACO, the open-source code of TACO does not include the implementation for MetaWorld and our attempts to contact the corresponding author have been unsuccessful. Despite our substantial efforts, we have not yet succeeded in reproducing expected performance.

Currently, although we have obtained some experimental results that are highly consistent and coherent with our previous conclusions, due to time constraints and limited computational resources, we are temporarily unable to include all the additional results we intended to supplement in the manuscript. Therefore, in order to make the paper more robust, enhance the comprehensiveness of the experiments, and further increase its potential impact, we have decided to withdraw it for more thorough improvement.

Lastly, we would like to thank all the reviewers for their valuable time and thoughtful feedback.

---

### Note · Authors · 2024-12-02

**Comment:**

We have decided to withdraw the paper for more thorough improvements to enhance the comprehensiveness of the experiments and increase its potential impact.

**Withdrawal Confirmation:**

I have read and agree with the venue's withdrawal policy on behalf of myself and my co-authors.